# Multiple In-Mold Sensors for Quality and Process Control in Injection Molding

**DOI:** 10.3390/s23031735

**Published:** 2023-02-03

**Authors:** Richárd Dominik Párizs, Dániel Török, Tatyana Ageyeva, József Gábor Kovács

**Affiliations:** 1Department of Polymer Engineering, Faculty of Mechanical Engineering, Budapest University of Technology and Economics, Műegyetem rkp. 3., H-1111 Budapest, Hungary; 2MTA-BME Lendület Lightweight Polymer Composites Research Group, Műegyetem rkp. 3., H-1111 Budapest, Hungary

**Keywords:** injection molding, pressure sensors, quality control, in-mold measurement

## Abstract

The simultaneous improvement of injection molding process efficiency and product quality, as required by Industry 4.0, is a complex, non-trivial task that requires a comprehensive approach, which involves a combination of sensoring and information techniques. In this study, we investigated the suitability of in-mold pressure sensors to control the injection molding process in multi-cavity molds. We have conducted several experiments to show how to optimize the clamping force, switchover, or holding time by measuring only pressure in a multi-cavity mold. The results show that the pressure curves and the pressure integral are suitable for determining optimal clamping force. We also proved that in-channel sensors could be effectively used for a pressure-controlled SWOP. In the volume-controlled method, only the sensors in the cavity were capable of correctly detecting the end of the filling. We proposed a method to optimize the holding phase. In this method, we first determined the integration time of the area under the pressure curve and then performed a model fit using the relationship between the pressure integral and product mass. The saturation curve fitted to the pressure data can easily determine the gate freeze-off time from pressure measurements.

## 1. Introduction

Injection molding is one of the most common processing methods in the plastics industry [1]. Although injection molding is considered a mature technology today, Industry 4.0 requires continuous improvement in process efficiency and part quality [2]. Numerous solutions have been developed to improve the economic feasibility of injection molding. One of the most effective solutions is the use of a multi-cavity mold. However, quality control for parts produced with a multi-cavity mold is difficult. First, injection molding is a complex process, and the quality of the resulting part is affected by numerous processing parameters, which can vary widely and are also interrelated. Second, the term “product quality” can be understood in various ways. We can distinguish the three most common groups of the quality indicators of injection molded products: dimensional and weight stability of manufactured parts [3], surface properties (roughness, sink marks, weld lines, etc.) [4], and physical properties (mechanical, optical, electrical, etc.) [5].

In most cases, the quality of a part is understood as a combination of the above criteria. A global trend in manufacturing is to relate the quality of parts with process parameters [6]. However, multiple process parameters of injection molding can vary in a wide range, and moreover, they interrelate in a complex way and, consequently, sometimes have a non-obvious influence on the quality of parts. Therefore, finding a relation between product quality and process parameters is a non-trivial task that requires a comprehensive approach, which often involves a combination of sensoring and information techniques [6]. Statistical analysis [7,8] and artificial intelligence (AI) [9], especially machine learning (ML) [10,11,12], are used more and more frequently today for the process optimization and quality control of industrial manufacturing processes. However, these methods are data-driven and often do not consider the physical aspects of injection molding. Therefore, it is essential to develop a process optimization and control technique that considers the physical processes and transformations that occur during injection molding.

Several studies have proven that weight is a reliable indicator for characterizing and controlling the quality of injection molding products and process stability, as variation in weight is inversely proportional to part quality [13,14,15,16]. The weight of injection-molded parts varies due to several reasons. One of the reasons is the differences in the specific volume of the melt caused by the inevitable variations of the injection molding process [15]. The problem of weight variation gets even worse in the case of a multi-cavity mold. First, the dimensions of individual cavities are not identical and can differ within tolerance limits. Second, the properties of a polymer melt vary from cavity to cavity due to differences in the temperatures induced by shear heating. Next to the wall, the shear rate is higher than in the middle, and more heat is generated in the melt. When the melt is divided in the runner system, the melt properties can be different in the branches, which leads to different part quality in various cavities. This phenomenon is called the imbalance of the runner system [17,18].

The direct measurements of part weight are usually implemented as a quality control procedure. However, it is essential to find a reliable process parameter with which part weight can be monitored and predicted online. Changes in polymer properties, particularly in the specific volume of the melt, indicate changes in the weight of the parts produced. Therefore, monitoring the specific volume of the melt through the measuring of pressure and temperature in the cavity of a mold is a reliable tool for predicting weight variation. According to Zhou et al. [15], the specific volume of the melt is mainly affected by pressure. Therefore, the authors proposed a pressure integral as an effective process parameter to predict the weight variations of parts and characterize their quality. The most relevant melt pressure data come from the runner and the cavity [19,20]. Consequently, monitoring pressure in the cavity and runner system is an excellent solution to collect real-time data, control the quality of injection molded parts [19], as well as test the injection mold itself [21,22]. Kemmetmuller et al. [23] proposed a method to estimate part mass using pressure and temperature sensors installed in the cavity of an S-shaped test mold.

A typical injection molding pressure profile includes the filling and the holding stage. The time at which the filling stage ends and the holding stage starts is referred to as the switchover point (SWOP). The right setting of the SWOP is crucial for the stability of the quality of injection-molded parts [24]. The traditional method to determine the SWOP uses an offline mass measurement. Such a method is very time-consuming and often leads to variations in the quality of the parts produced. Huang [25] proposed a method based on the cavity pressure data and a grey prediction model. Chen et al. [26] proposed a simple least-squares regression-based approach for determining the appropriate switchover point.

Numerous studies demonstrate the importance of the holding phase for decreasing shrinkage and warpage in injection molding parts [27,28,29,30,31]. Increasing holding pressure reduces shrinkage [29,30]. The influence of holding pressure on warpage is a controversial topic. For example, Li et al. [32] demonstrated that higher holding pressure reduces warpage, while Barghash and Alkaabneh [33] stated the opposite. Simulations showed that increasing holding time could cause higher warpage. Therefore, it is essential to find optimal parameters (mainly holding time and holding pressure) for the holding phase. The end of the holding phase is indicated by a gate freeze-off, as after the solidification of the gate, no material can enter the mold. Consequently, to obtain the shortest cycle time, it is of paramount importance to predict gate freeze-off time as accurately as possible, which is not a trivial task. The most common way to determine gate freeze-off time is simply using a long holding time to ensure that the gate freezes [34]. However, we found several prediction methods in the literature. For example, Leo and Cuvelliez [35] identified a gate freeze-off time from the pressure curves obtained with the pressure transducers installed in the cavity. The same authors also mentioned that gate freeze-off time can be identified from the packing time–part weight curve. They noted that the leveling off of weight curves indicates the freezing of the gates. Pantani et al. [36] and De Santis et al. [27] also evaluated gate freeze-off time by monitoring the weight of the parts produced with increasing holding time.

The clamping force is another vital processing parameter that influences the quality of a part. Inappropriate settings of the clamping force can result in dimensional inaccuracies and even flash in the parts produced. This issue is crucial for low-viscosity polymers, as flash defects may occur under even a very small opening of the mold halves during the molding cycle. A common practice to overcome this issue is setting the upper limit of the clamping force as default. However, such an approach shortens the lifetime of both the injection molding machine and the mold and increases energy consumption. Therefore, it is essential to determine the optimal clamping force. Traditional methods of estimating the optimal clamping force mainly use the total projected area of the cavity, sprue, and runner along the clamping direction multiplied by the predicted cavity pressure of the molten polymer. However, this prediction is quite rough. Huang et al. [37] proposed a method to define an optimal clamping force based on the characteristics extracted from the tie bar elongation profile under different clamping force settings and regression analysis of these data points. The proposed methodology accurately determines an optimal clamping force using just six shots. Yang et al. [38] proposed an experimental method to determine the optimal clamping force by measuring the difference between the clamping force before mold filling and after cooling. The optimal clamping force, according to the authors, corresponds to the moment when the clamping force change becomes zero.

There is a lot of research dedicated to the monitoring of the quality, control, and optimization of injection molding processing parameters. However, we have not found a methodology that effectively helps predict the quality of the parts produced in a multi-cavity mold. In this research, we investigated the suitability of pressure sensors for controlling the injection molding cycle. We conducted several experiments to optimize the clamping force, the SWOP, or holding time by measuring only pressure in a multi-cavity mold. We also illustrated the importance of pressure sensors in an experiment where we investigated the effect of the injection rate and melt temperature on mold filling imbalance.

## 2. Machines, Materials, and Methods

The samples were molded with an Arburg Allrounder 420 C 1000-290 (Arburg Gmbh+Co, Loßburg, Germany) injection molding machine from acrylonitrile butadiene styrene (ABS), named Terluran GP-35 (INEOS Styrolution, Manchester, UK). Table 1 shows the main process parameters (recommended by the manufacturer) and the mechanical properties of this material. We used a 16-cavity mold with 34 built-in pressure sensors (Figure 1). The pressure sensors were PC 15-1-AA indirect piezoelectric sensors from Cavity Eye (Cavity Eye Ltd., Kecskemét, Hungary), installed behind the ejector pins. Product weight from the first, third, and fifth experiments was measured with an Ohaus Explorer analytical balance (OHAUS Europe Gmbh, Uster, Switzerland). We performed five experiments to demonstrate the suitability of pressure sensors in controlling multi-cavity molds and to illustrate filling imbalance. The settings for each injection molding experiment are summarized in Table 2 and Table 3. Table 2 shows the fixed parameter set, while Table 3 contains the variable parameters and their values. The mathematical evaluation and statistical analysis were performed with the Matlab R2021 (The MathWorks Inc., Natick, MA, USA) software package.

In the first experiment (01—clamping force), our goal was to optimize the clamping force based on mass and pressure measurements. For this experiment, we produced specimens with ten different clamping forces. For each setting, five cycles were sampled, where the mass of the products and the pressure in the mold were measured.

In the second (02—pressure-controlled SWOP), third (03—hybrid SWOP), and fourth (04—filling imbalance) experiments, we investigated the filling stage. The second experiment showed the advantages and disadvantages of switching based on in-mold pressure. In this experiment, the end of the filling stage was controlled using in-mold pressure. We used three different sensor locations and four different pressure thresholds to investigate the effect of these parameters on the SWOP. During this experiment, we did not use holding pressure so that we could investigate the pressure overflow in the mold. In this experiment, we sampled 15 pressure curves for each setting to make sure the switchover was stable.

The third experiment was a hybrid method of switching from the filling to the holding stage because the settings were changed manually. However, we compared the measured pressures with the measured mass to show the advantages of online measurement. Here, we injection-molded products with seven different switchover volumes. Three cycles were sampled, where the mass of the products (usually short shots) and the pressure in the mold were measured.

The fourth experiment measured mold filling imbalance and showed the dependence of this imbalance on melt temperature and injection rate. We used three different melt temperatures and seven injection velocities for this experiment and sampled in-mold pressure in five cycles for each setting.

Finally, we conducted the fifth experiment (05—gate freeze-off) to investigate the gate freeze-off time based on pressure measurement during the holding phase. Here, we injection-molded products with 13 different holding times. We sampled five cycles, where the mass of the products and the pressure in the mold were measured.

## 3. Results

### 3.1. Controlling the Clamping Stage

The clamping force should be high enough to prevent mold flash. However, if the clamping force is excessive, it can ruin the airflow in the cavity and lead to burn marks on the product. Moreover, too great a clamping force increases energy consumption unnecessarily. In Figure 2, we showed that maximum cavity pressure is significantly lower when we use a lower clamping force. Moreover, cavity pressure at a low clamping force decreases much more slowly than at a high clamping force. This is because the injected melt can open the mold due to insufficient clamping force, forming a flash on the edge of the product. When the machine uses the predefined force to keep the tool closed continuously during the cycle, it pushes the product to the sensor, which leads to a higher pressure and slower pressure decrease.

When the clamping force is increased, melt pressure cannot open the mold. Consequently, less melt gets in the cavity, and it compresses better. This can be tracked on the pressure curves, too. As the clamping force is increased, the difference between the in-mold pressure curves becomes smaller and smaller until it disappears. However, from the computational point of view, it is preferable to store only selected characteristic features of the pressure curve rather than the whole curve so that we can observe the difference between each set of parameters. Figure 3a shows the pressure integral of the post-gate sensors for the inner and outer cavities of the mold. The trend of part weight change (Figure 3b) is very similar to the trend of the pressure integral. For each cavity, the correlation coefficient is greater than 0.96 with a *p*-value smaller than 10^−8^, which corresponds to a strong correlation between the two values and a low probability of no significant linear correlation between the two values.

In order to make the trends quantifiably comparable, we carried out a one-way analysis of variance (one-way ANOVA). With this test, we check whether there is a significant difference between the resulting pressure integrals or product masses when the clamping force is varied. However, a one-way ANOVA alone cannot make pairwise comparisons, so it can only show if there is at least one outlier at different clamping forces. However, post hoc testing allows us to compare each clamping force pair by pair, and we can see if there is a significant difference between them. Therefore, for the ANOVA results, we performed a Tukey–Kramer post hoc test, which performs the pairwise comparison for each pair of clamping force settings. This test reduces the type I error rate by adjusting the calculated *p*-values. If the adjusted *p*-value is greater than 0.05 (significance level), then there is no significant difference between the compared values.

In Figure 4, we collected the adjusted *p*-values for each pair of clamping forces with the metrics mentioned earlier (part weight and pressure integral). Therefore, Figure 4 is effectively two parts separated by the black diagonal. As shown in Figure 4, if one of the compared clamping force pairs is small, the *p*-value is almost zero (deep blue squares), i.e., the difference between the masses/pressure integrals of the two products produced at different clamping forces is significant. On the other hand, when the compared closing forces are large, the *p*-values are large, so there is no significant difference between the measured masses or pressure integrals. As the two parts on either side of the diagonal are similar, we can say that both metrics can be used in a similar way to set the clamping force, although pressure measurement takes considerably less time for a multi-cavity mold.

### 3.2. Controlling the Filling Phase

The second and third experiments compared different control methods for the end of the filling stage. For the multi-cavity mold used, we investigated which sensors in the mold were suitable for controlling the switching based on the length of the runner from the sprue.

During the filling stage, the melt is injected from the barrel through the runner system into the cavity with a relatively high velocity. Melt pressure decreases during the path of the melt to the cavity. This raises the question: “Which pressure sensor location and which pressure threshold should be set as a switching control in the mold?” Theoretically, a small pressure signal from a sensor installed at the end of the cavity is enough to switch between the holding and filling stages. However, feedback from a sensor has a delay, which can cause higher in-mold pressure than necessary. We examined three different locations of pressure sensors along the melt path to determine the effect of this delay (see Figure 5). We used pressure sensors at a tertiary runner, after the gate of a cavity (post-gate sensor), and at the end of a cavity. We set the switchover pressure threshold to 50 bar. Still, due to the sampling rate of the sensor (100 Hz), the feedback signal was sent much later, especially for sensors towards the end of the flow path, and the machine also had a delay time. The solid lines show the pressure curves until the switchover, and the dashed lines show the in-mold pressure afterward. It is clear that even though there was no holding pressure, in-mold pressure increased significantly after the switchover if sensors inside the cavity controlled the machine. On the other hand, the sensor in the runner was able to take the pressure off. However, the melt did not reach the sensor at the end of the cavity, which meant an incomplete product.

Generally, the desired transition between the filling and holding stage is such that there is no sudden significant change in pressure due to the switchover point. This means that after the cavity is filled, there is a short time when the melt is compressed during filling. After that, when holding pressure is applied, the additional melt is pushed into the cavity to decrease the shrinkage of the product. If the pressure drops during this transition, it means that too much pressure is applied during filling or the holding pressure is insufficient. In the former case, the mold may open (if the clamping force is not high enough), and a flash can appear at the side of the product. If injection pressure is too low, the product will contain sink marks due to the lack of material in the thicker parts of the product. To get a smooth switchover on the pressure curve, we examined the effect of switchover pressure on the data obtained with the sensor installed at the tertiary runner. Figure 6 shows how in-mold pressure changes when the pressure trigger is 50 bar, 100 bar, 125 bar, or 150 bar. With a 50-bar trigger level, the melt did not reach the cavity; in the case of 100 bar, the melt reached the end of the cavity sensor. However, the pressure in the cavity was below 70 bar, which means the cavity was only filled, but there was no compression during the filling stage. In the case of a 125-bar or 150-bar trigger, we found that in-mold pressure increased significantly after the SWOP, even though we did not apply holding pressure. This phenomenon is most probably caused by the accumulated melt in the runner system. When the melt reaches the sensor, the pressure increases slightly until the melt comes to the small-diameter gates, where higher pressure is needed to get the melt into the cavity. After the cavities are filled, the slope of the pressure curve changes dramatically because of the compression of the melt. Melt pressure reaches the switchover limit, but due to the delay in control, the screw moves forward, pushing the melt into the mold. The main goal is to find the processing window to control the filling stage through pressure sensors. The further we place the sensor from the end of the cavity, the less accurately we can determine the end of the filling stage.

### 3.3. Manual Switchover Method with Pressure Measurement

We used a different approach for the third experiment to determine the SWOP. Here we defined the SWOP with the melt volume limit and performed a switchover volume sweep. The dose volume prepared for the next injection cycle was not changed, only the switchover volume. Therefore, the injected melt volume changed. During injection molding, the in-mold pressure was measured, and three samples were taken from each cavity at each setting. We did not use holding in this test, so the weight of the parts shows how each cavity was filled based on how much polymer melt was injected into the mold. In Figure 7, the change in part weight can be seen due to the change in the injected melt volume. Mass measurement is a traditional method to define the switchover volume. This method is robust but relatively time-consuming, especially in the case of multi-cavity molds. Therefore, in this study, we developed another prediction method, which is based on cavity pressure data. To verify the pressure-based method, we compared the obtained results with those delivered by the mass measurement method. Therefore, we investigated the variation of the maximum pressure at different sensors along the flow path as a function of the amount of melt injected in the mold (Figure 8).

The mass measurement results (Figure 7) indicate that there is a significant difference in the filling between cavities 1 and 2. Subsequent gate measurements show that the gate at cavity 2 is about one and a half times the size of the other gates. This means a much lower resistance during filling, which results in a significantly greater product weight. From the mass measurement results (Figure 7), it is visible that when 22 cm^3^ of melt was injected, cavity 08 was already filled, while cavity 07 was almost filled. The sensors at tertiary channels cannot show the difference between the cavities connected to that channel (Figure 8a), which is a great disadvantage. The sensors in front of the gate can separate the individual cavities due to their position, but they cannot show the filling of the cavities either (Figure 8b). In contrast, the pressure data obtained from the sensors in the cavities clearly show that the melt flows differently in the individual cavities. The maximum pressure measured at the end of the cavity sensor shows this filling phenomenon the best because the maximum pressure in cavity 08 is already greater than zero when the volume of the injected melt is 22 cm^3^ (Figure 8d). The results of the post-gate sensors show a similar trend (Figure 8c). Figure 8c,d show that cavities 02, 05, and 06 are filled next due to the almost 100 bar maximum pressure jump. Cavities 01, 03, and 04 were the last to be filled when 24.5 cm^3^ of melt was injected into the mold. The sensors in the channel can show the filling of these last cavities as the slope of the maximum pressure function changes significantly from here. However, the in-cavity sensors demonstrate higher accuracy in this case. Therefore, in-cavity sensors are much more recommended for this method, while their use was less promising in the previous method.

### 3.4. Mold Filling Imbalance Detection with a Pressure Sensor

As was demonstrated before, there is a clear mold-filling imbalance (Figure 7 and Figure 8). However, the methods shown before the results highlighted a way of measuring the dependence of mold filling imbalance during filling on melt temperature and injection rate. Therefore, we experimented with three melt temperatures and seven injection velocities to see if the effect of these factors is visible based on pressure measurement. The end-of-cavity sensors were used to measure mold filling imbalance, as the previous experiment showed that these were particularly suitable for monitoring the filling phase. The measurement procedure was as follows: from the pressure curve, we defined the time of arrival of the melt to each cavity by checking when the pressure exceeded 5 bar. Then the ratio of the longest and shortest melt arrival times was calculated for each injection rate–melt temperature combination for each side of the mold. The shortest arrival times were found for the inner cavities (cavities 07 and 08), while the longest times were for cavities 03 or 04 for both low and high speeds (Figure 9). This ratio was used to analyze the filling imbalance.
(1)$$t_{ratio}=\frac{\max\left(t_{arrival}\right)}{\min\left(t_{arrival}\right)}$$
where $t_{ratio}$ is the ratio of the longest and shortest melt arriving time on the specified side of the mold, $t_{arrival}$ is the melt arrival time for each cavity for the specified side of the mold.

We analyzed whether melt temperature and injection rate significantly affected the filling imbalance (Table A1 and Table A2 in Appendix A). The analysis clearly showed (with a significance level of 0.05) that both factors strongly affected the imbalance metric, and there was considerable interaction between the two factors (*p*-values were much smaller than 0.05). This means that the effect of the injection rate highly depends on the level of melt temperature (see Figure 10). As the injection rate increases, the ratio deviates further from the optimum value of 1, but this effect can be reduced by increasing melt temperature.

### 3.5. Controlling the Holding Phase

The holding phase aims to reduce part shrinkage caused by the change in specific volume during the cooling phase. During the holding phase, the machine injects more material into the mold, typically with pressure control. After the filling phase, only a small amount of melt is needed to compensate for the change in specific volume. One of the crucial parameters is holding time, which should be equal to gate freeze-off time in an optimal case. If the holding time is shorter than the gate freeze-off time, there is insufficient compensation, which means higher shrinkage. However, if the holding phase continues after the gate is frozen, the longer cycle time causes a loss of income, and we put more material into the runner system, which can increase the amount of waste. A well-established method to determine gate freeze-off time is to measure the weight of parts molded with different holding times. However, this method could take a long time, especially in the case of multi-cavity molds. Automation with robots could reduce this time, but this would be an expensive solution. Therefore, a process may be needed to determine gate freeze-off time.

In our last experiment, we injection-molded parts in the mold with different holding times in the range of 0–3 s with a step of 0.25 s. In each setting, we collected five samples and measured the in-mold pressure in different positions during the flow path. Figure 11 shows how the pressure curve changes as a function of the holding time in two different locations (one before the gate in the runner and one after the gate in the cavity). It is clear that if the holding time increases, the area under the pressure curve also increases in the runner. However, the pressure approaches a limit in the cavity, and the area does not increase after this limit. After the gate freezes, the pressure cannot be maintained because the frozen gate closes the cavity. Therefore, measuring the cavity pressure integral can be an excellent way to detect gate freeze-off time because this measurement can be done online during production.

To verify this hypothesis, we measured the mass of the products and compared them with the results of the calculation of the pressure integral. However, the time of pressure measurement depends on cycle time and the change in pressure in the cavity. If the sampling time is too short, meaningful information about the decay of the pressure curve may be lost. If the sampling time is too long, the cycle time may be exceeded. Therefore, we examined the effect of the integration interval on the relation between mass measurement and pressure integrals. We calculated the integral with different interval lengths. The shortest interval was 0.2 s, the longest interval was 10 s from the start of measurement, and the step size was 0.2 s. We calculated the Pearson correlation coefficient for each pressure integral–weight relationship in the case of different intervals and the corresponding *p*-value for each cavity. Figure 12 shows the estimated coefficients and *p*-values as a function of the end of the integration time. The trend of the values was very similar for all cavities. These results show that the correlation coefficient clearly indicates a strong positive correlation between the pressure integral and part weight. We showed the 95% confidence interval for each correlation coefficient and marked the adjusted *p*-values for each correlation coefficient. The adjusted *p*-values were needed because the experiment involved multiple comparisons, which increased the likelihood of the type I error rate. Therefore, the Bonferroni-Holm correction was applied to the *p*-values from the correlation. The size of the confidence intervals and the *p*-values indicate that the assumed relationship is significant at the integration time of 3 s.

Therefore, the integral was calculated from the start of the measurement until the third second. We determined the Pearson correlation coefficient between the mass of each sample in the first eight cavities and the pressure integral measured from these cavities. The correlation coefficient is R = 0.973 with a significance level of α = 0.05 (Figure 13).

Figure 14 shows the change in product mass and the pressure integral as a function of holding time. We used the following saturation curve function to estimate the change of the pressure integral or mass:(2)$$y\left(t_{hold}\right)=y_{0}+\left(y_{\infty }-y_{0}\right)\cdot \left(1-e^{-\tau \cdot t_{hold}}\right)$$
where $y\left(t_{hold}\right)$ is the mass or the pressure integral as a function of holding time, $y_{0}$ is the mass or pressure integral without a holding pressure, $y_{\infty }$ is the mass or pressure integral after the gate is frozen, τ is the time parameter, and $t_{hold}$ is the holding time. The component $\left(y_{\infty }-y_{0}\right)$ can show how much compensation the holding phase can make until the gate freezes. On the other hand, τ defines the steepness of the function, which shows how quickly the gate freezes. After we determined the constants from the fitting, we set a threshold, which we used to calculate the gate freeze-off time (Figure 15). If the pressure integral function or mass function crossed this threshold, we could assume that the gate is frozen because of the relatively small change in the curves.

This method can accurately predict gate freeze-off time. We examined the gate freeze-off times, and nearly all cavities froze after approximately 2 s of holding time (Figure 15). This moment (gate freeze-off) occurred at the third second of the pressure curve, which explains why the *p*-value of the correlation between the pressure integral and product weight was the smallest when integration was performed until the third second (Figure 11 and Figure 12).

## 4. Conclusions

We investigated several methods of using pressure sensors to control multi-cavity molds. One such method was to optimize the clamping force. The results show that the pressure curves and the pressure integral are suitable for determining the optimal clamping force. This method can save time and energy, as we can use the information from the pressure curves to find the optimal clamping force faster, and its value may be lower than the maximum clamping force of the machine.Subsequently, we compared two methods for controlling filling as a function of in-mold sensor location. The results showed that the use of in-channel sensors is recommended for a pressure-controlled SWOP. In contrast, in the volume-controlled hybrid method, the sensors in the cavity were the only sensors capable of correctly detecting the end of filling.The dependence of mold filling imbalance on injection rate and melt temperature was examined with in-mold sensors. Our results show that the imbalance increases with the injection rate, but this effect can be reduced by increasing the temperature of the melt.In the last experiment, we optimized the holding phase. We first determined the integration time of the area under the pressure curve and then performed a model fit using the relationship between the pressure integral and product mass. The saturation curve fitted to the pressure data can easily determine gate freeze-off time from pressure measurements. There was little difference between the gate freeze-off times calculated from mass measurements and pressure measurements.

## Figures and Tables

**Figure 1 sensors-23-01735-f001:**
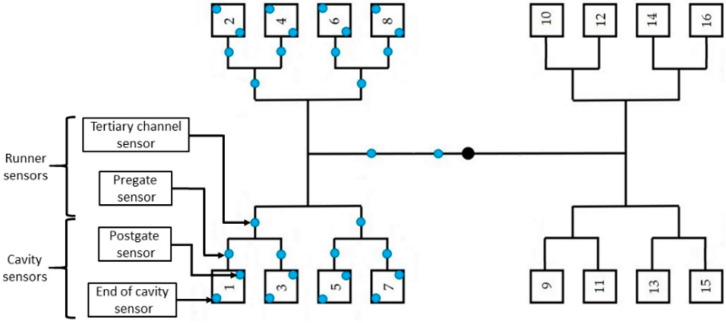
A multi-cavity mold and locations of pressure sensor locations.

**Figure 2 sensors-23-01735-f002:**
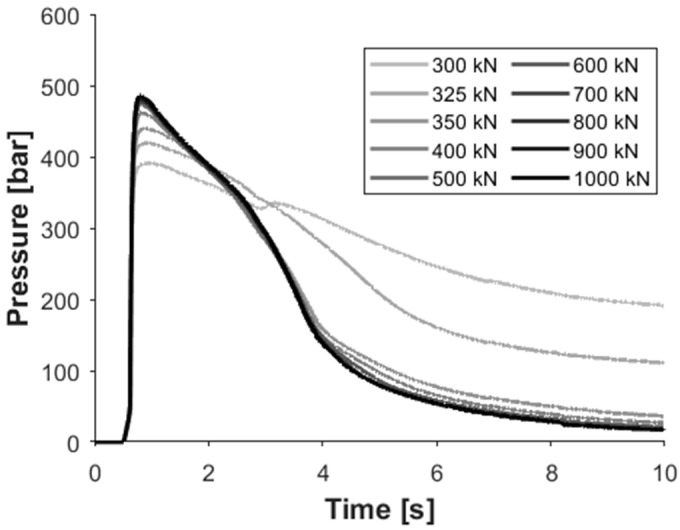
In-mold pressure change as a function of time at different clamping forces (cavity 04, post-gate sensor).

**Figure 3 sensors-23-01735-f003:**
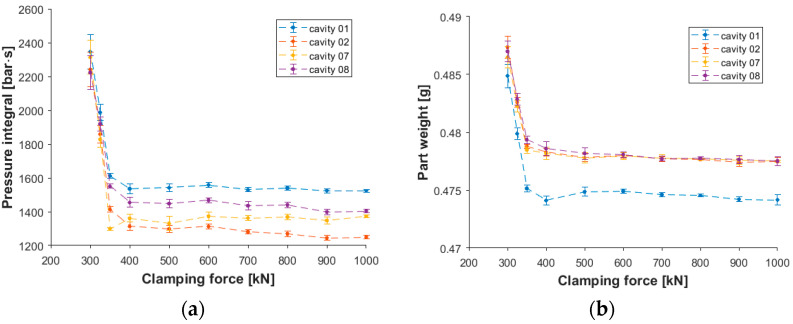
The change of the pressure integral (**a**) and part weight (**b**) of the inner (07 and 08) and outer (01 and 02) cavities due to a change in the clamping force.

**Figure 4 sensors-23-01735-f004:**
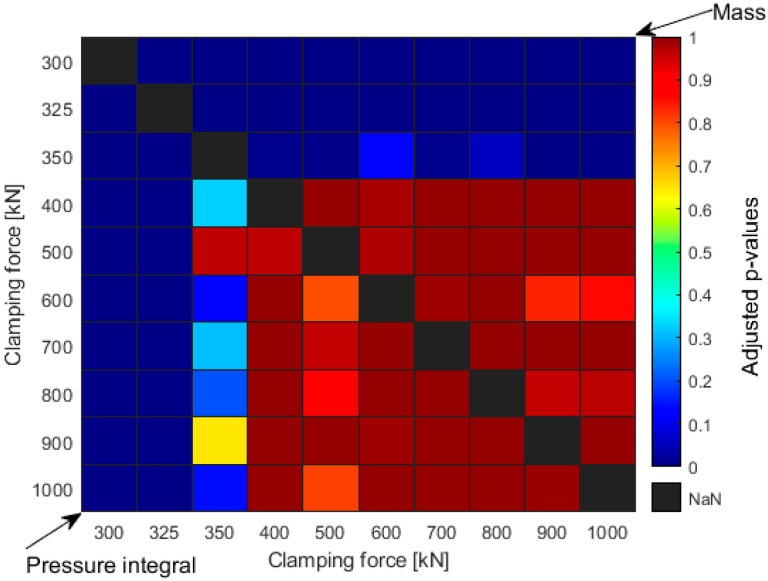
The adjusted *p*-values (from the Tukey–Kramer post hoc test) for each clamping force pair for the pressure integrals (below the diagonal) and part weights values (above the diagonal).

**Figure 5 sensors-23-01735-f005:**
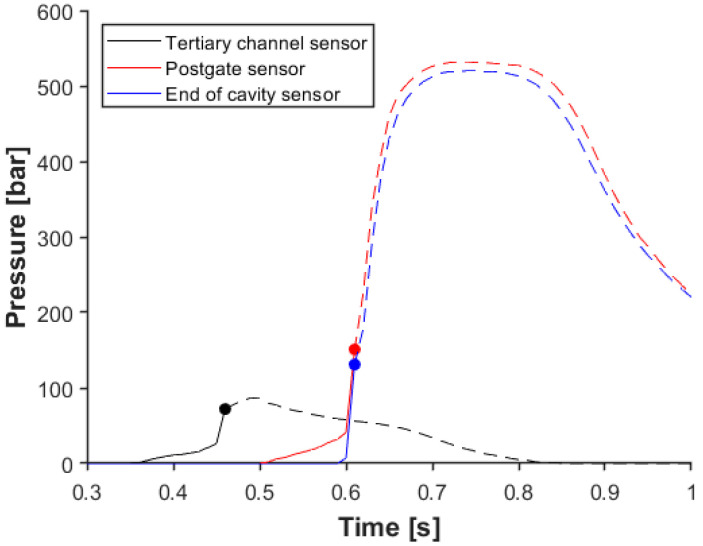
The effect of sensor location on switchover pressure (sensors from the flow path to cavity 02).

**Figure 6 sensors-23-01735-f006:**
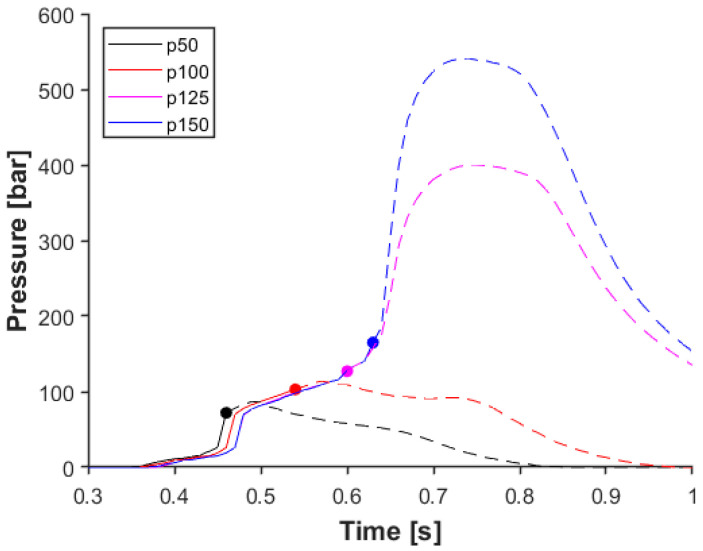
The effect of the switchover pressure threshold on in-mold pressure, measured with a sensor in a runner (tertiary channel sensor to cavity 02).

**Figure 7 sensors-23-01735-f007:**
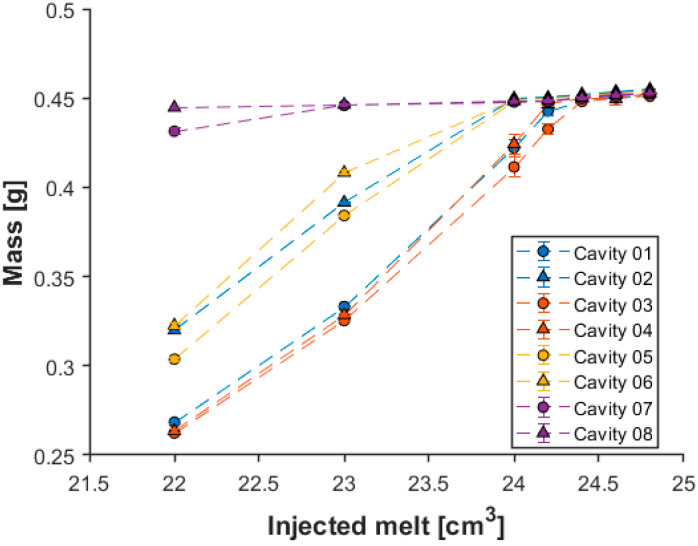
Change of part weight due to the injected melt volume.

**Figure 8 sensors-23-01735-f008:**
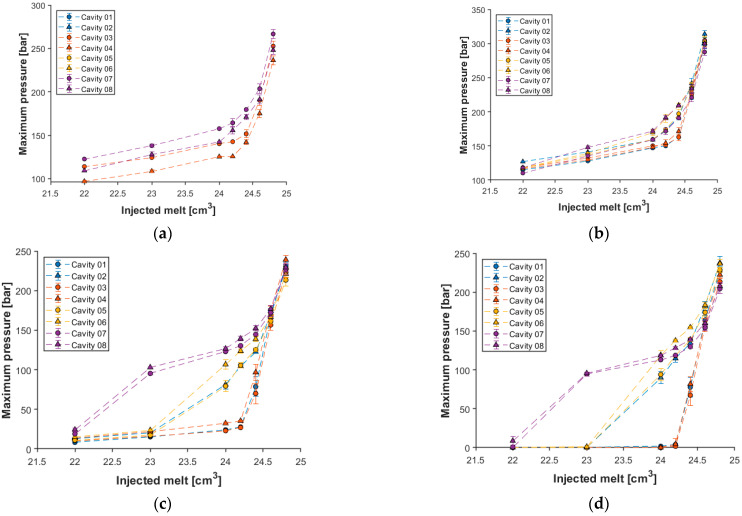
Maximum pressure for each cavity based on: (**a**) tertiary channel sensors; (**b**) pre-gate sensors; (**c**) post-gate sensors; (**d**) end-of-cavity sensors.

**Figure 9 sensors-23-01735-f009:**
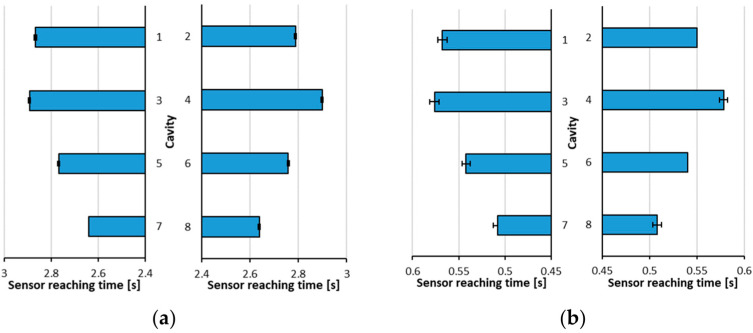
Melt arrival times to the end-of-cavity sensors: (**a**) v_inj_ = 10 cm^3^/s and for (**b**) v_inj_ =110 cm^3^/s (*T*_melt_ = 215 °C).

**Figure 10 sensors-23-01735-f010:**
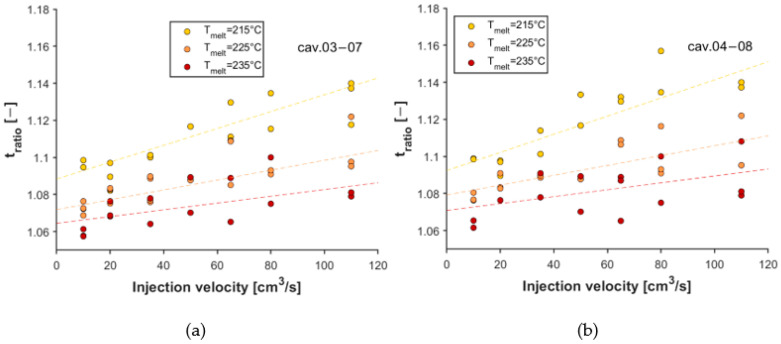
Effect of melt temperature and injection rate on the ratio of the longest and shortest melt arrival times: (**a**) cavity 03–07; (**b**) cavity 04–08.

**Figure 11 sensors-23-01735-f011:**
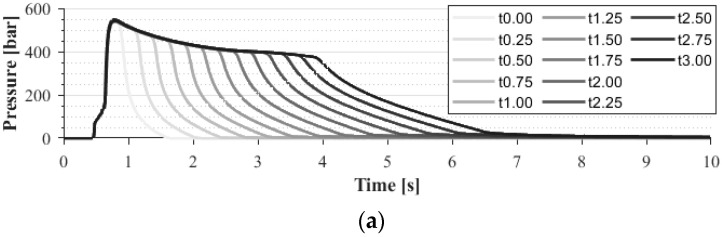
In-mold pressure curves as a function of time and holding time measured with a: (**a**) pre-gate sensor; (**b**) post-gate sensor (cavity 01).

**Figure 12 sensors-23-01735-f012:**
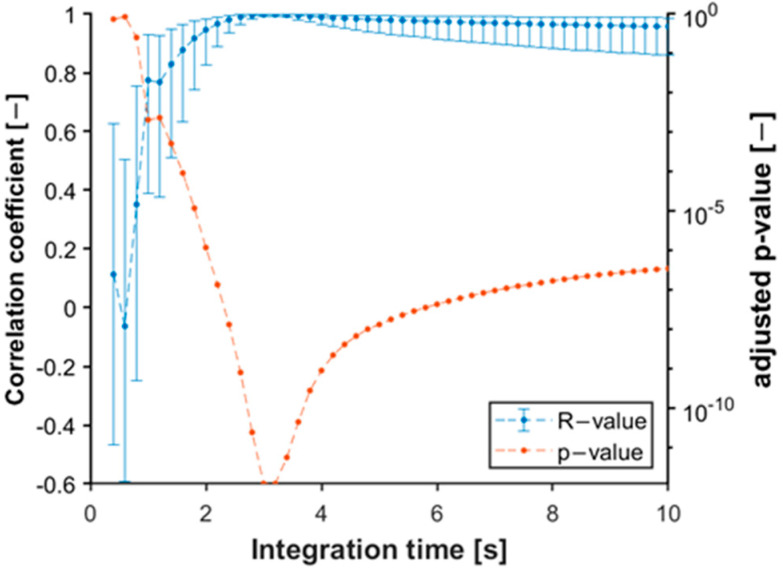
The changing of the correlation coefficients and their 95% confidence interval between the pressure integral and mass based on integration time and the adjusted probabilities for each correlation (cavity 08).

**Figure 13 sensors-23-01735-f013:**
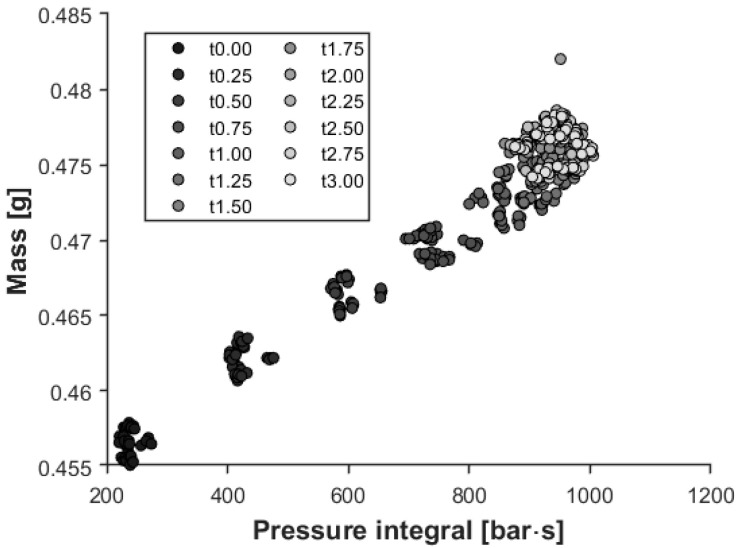
Relationship between the pressure integral and sample mass (R = 0.973 with a significance level of 0.05).

**Figure 14 sensors-23-01735-f014:**
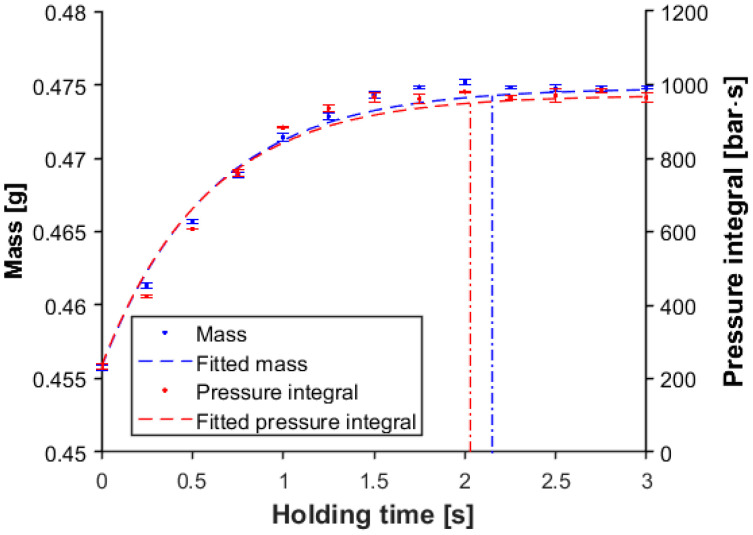
Mass and pressure integral as a function of holding time (cavity no. 1, post-gate sensor, p_hold_ = 600 bar, integration time 3 s).

**Figure 15 sensors-23-01735-f015:**
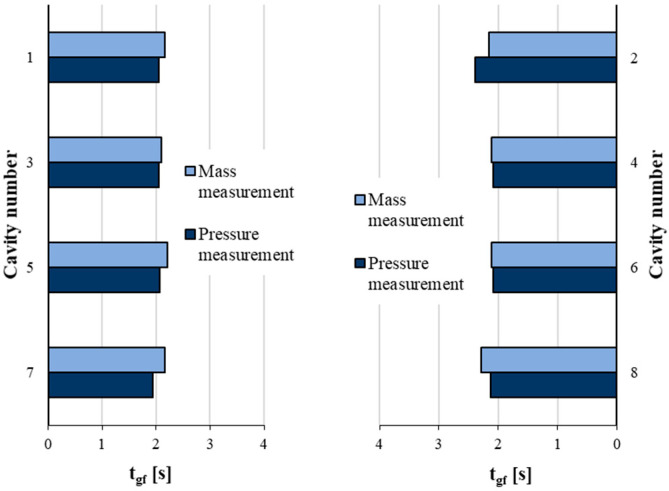
Predicting gate freeze-off time from the saturation curve from the measurement of mass and pressure.

**Table 1 sensors-23-01735-t001:** Recommended processing parameters by the manufacturer (Terluran GP-35 ABS) and the mechanical properties of the material.

Processing Parameter	Values
Drying temperature and time	80 °C for 4 h
Recommended melt temperature range	220–280 °C
Recommended mold temperature range	30–60 °C
Mechanical Properties	Values
Tensile stress at yield at 23 °C	44 MPa
Tensile strain at yield at 23 °C	2.4%
Charpy notched impact strength at 23 °C	19 kJ/m^2^

**Table 2 sensors-23-01735-t002:** Injection molding parameters—fixed parameter set.

	Values				
Process parameter	Exp. 01	Exp. 02	Exp. 03	Exp. 04	Exp. 05
Clamping force, kN	-	700	700	700	700
Injection rate, cm^3^/s	50	50	50	-	50
Switchover control	Volume	Pressure	Volume	Volume	Volume
Switchover point, cm^3^	7	-	-	6	7
Screw rotation speed, m/min	15	15	15	15	15
Back pressure, bar	40	40	40	40	40
Decompression, cm^3^	5	5	5	5	5
Dose volume, cm^3^	26	26	26	26	26
Holding pressure, bar	600	0	0	600	600
Holding time, s	2	0	0	2	-
Cooling time, s	15	18	18	15	18
Melt temperature, °C	225	225	225	-	225
Mold temperature, °C	40	40	40	40	40

**Table 3 sensors-23-01735-t003:** Injection molding parameters—variable parameter set.

Experiment Number	Changed Setting	Setting Levels
01—clamping force	Clamping force, kN	300/325/350/400/500/600/700/800/900/1000
02—pressure controlled SWOP	Switchover pressure limit on sensors, bar	50/100/125/150
03—hybrid SWOP	Switch over volume, cm^3^	9.0/8.0/7.0/6.8/6.6/6.4/6.2
04—imbalance	Melt temperature, °C	215/225/235
	injection rate, cm^3^/s	10/20/35/50/65/80/110
05—gate freeze-off	Holding time, s	0.00/0.25/0.50/0.75/1.00/1.25/1.50/ 1.75/2.00/2.25/2.50/2.75/3.00

## Data Availability

The data presented in this study are available on request from the corresponding author.

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
