# Peer review of "Multiple In-Mold Sensors for Quality and Process Control in Injection Molding"

_sensors, 2023, doi:10.3390/s23031735_

Round 1
Reviewer 1 Report
The authors integrated sensors to monitor injection molding process. The following issues should be addressed.
1. The images of equipment setup should be provided.
2. The pressure of materials flowing in the cavity should be monitored and it might be more important to determine the quality of the injection.
Reviewer 2 Report
The article submitted by Parizs. And Kovacs. et al, entitled “Multiple in-mold sensors for quality and process control in injection molding” (sensors-2181406), investigated the feasibility of controlling multi-cavity molds using pressure sensors. Although the motivation is good, however, in this reviewer’s opinion, the work is somewhat incomplete. For the benefit of the reader, a revision is suggested before its publication.
After reading all the paper, the method was introduced clearly. However, what is the meaning of the article should be introduced carefully. As a research article, this article is somewhat talking about the topic in general terms.
More examples using the pressure sensors to test/know the mold should be introduced. For example, Polymers for Advanced Technologies, 2020, 31, 2136, the foaming status was tested using the pressure sensor, authors may decide to discuss it or not by themselves.
Further comparison is necessary. For example, in figure 3, some results were listed. However, these are only original results.
In conclusion, some revisions are suggested before its publication.
Reviewer 3 Report
In this work, a process optimization and control technique that considers the physical processes and transformations that occur during injection molding was developed. The authors illustrated the importance of pressure sensors in an experiment, where they investigated the effect of injection rate and melt temperature on mold filling imbalance. The results provided several methods of using pressure sensors to control multi-cavity molds.
Some detailed comments:
*It is recommended that the authors give a table of performance parameters or references for injection materials.
*The different curves in Figure 2 and the different points in Figure 12 cannot be clearly distinguished.
*The abstract is too short and should describe in more detail - the background of this study, which questions were addressed, which research methods were used, the conclusions obtained, and the significance of this study.
*In Figure 6, why is there a significant difference between the curves of cavity 1 and cavity 2 when there is little difference between the curves of the other corresponding combinations (3-4, 5-6, 7-8)?
In conclusion, this article is recommended to be minor revised.
Reviewer 4 Report
A fresh view about merging sensor and information in manufacturing. The idea is in the main current developments, as it was defined in Tool wear monitoring of high-speed broaching process with carbide tools to reduce production errors, Mechanical Systems and Signal Processing 172, 109003 that can help you with initial descriptions. In addition the idea of using machine learning and boosting ensemble can help you as well, as it is shown in https://doi.org/10.1016/j.jmsy.2018.06.004
Conclusions are difficult to be read, it would be better as points, one per each highlight.
What is This ratioОшибка! Источник 295 ссылки не найден. was used to analyze filling imbalance…I 5think is a typo from other language.
You control the filling phase, using pressure. Did you cjeck temperature. Why not?
Section 2 is OK. I like the way to present results as well.
How many experiments and repetitions did you use. I think 3 to 8 are required, after reading works about the topic.
My only concern is the state of the art. Current works in manufacturing and a whole trend in many countries is to relate quality with process parameters, and you did not defend the idea in a proper way. Check above works and extend the discussion.
Round 2
Reviewer 1 Report
The revision is satisfying.
Reviewer 2 Report
The revised version is better than before. I recommend its publication in present form.
Reviewer 4 Report
Paper is OK, follow this line in future works: sound state of the art, new ideas, that is OK.